# Auxiliary Decision-Making System for Steel Plate Cold Straightening Based on Multi-Machine Learning Competition Strategies

**Zhen-Hu Dai** [1,2,†]**, Rui-Hua Wang** [3,†] **and Ji-Hong Guan** [1,*]

1 School of Electronic and Information Engineering, Tongji University, Shanghai 200070, China
2 Shanghai Baosight Software Co., Ltd., Shanghai 201203, China
3 School of Software, Nanchang University, Nanchang 330047, China
* Correspondence: jhguan@tongji.edu.cn; Tel.: +86-18616102875
† These authors contributed equally to this work.

**Abstract:** In the process of steel plate production, whether cold straightening is required is significant to reduce costs and improve product qualification rates. It is not effective by adopting classic machine learning judgment algorithms. Concerning the effectiveness of ensemble learning methods on improving traditional machine learning methods, a steel plate cold straightening auxiliary decision-making algorithm based on multiple machine learning competition strategies is proposed in this paper. The algorithm firstly adopts the rough set method to simplify the attributes of the conditional factors for affecting whether the steel plate cold straightening is required, and reduce the attribute dimensions of the steel plate cold straightening auxiliary decision-making data set. Secondly, the competition of training multiple different learners on the data set produces the optimal base classifier. Finally, the final classifier is generated by training weights on the optimal base classifier and combining it with a centralized strategy. While the hit rate of good products of the final classifier is 97.9%, the hit rate of defective products is 90.9%. As such, the accuracy rate is better than the single kind of simple machine learning algorithms, which effectively improves the product quality of steel plates in practical production applications.

**Keywords:** steel plate production; attribute reduction; ensemble learning; machine learning

## 1. Introduction

Currently, industrial real-time databases need AI data analysis technologies to solve the problems of defect detection [1], sorting identification [2], size detection [3], visual guidance [4], etc., which help enterprises to achieve flexible production [5] and high automation. For example, the intelligent defect detection makes the manual detection to be a laborious task as the human eye is not able to distinguish the fast moving tiny objects. Furthermore, the excessive use of eyes is much easier to miss objects. AI technologies can be used to overcome these difficulties and increase detection efficiency by means of adjusting the detection accuracy according to the product detection requirements. At the same time, AI technologies can realize automatic detection, automatic processing and reducing the rate of defective products and labor costs by means of cooperating with automatic production lines, which increase the production efficiency significantly [6].

However, the classical single AI algorithm applied in the current industrial production process is still insufficient. A single AI algorithm has different prediction capabilities for different data sets, and its adaptability to multiple production environments and multiple steel plates is also insufficient. It is also easier to fall into local optimization and cannot obtain the optimal solution.

Aiming at the shortcomings of classical single AI algorithm, this project proposes an ensemble learning algorithm based on multi-machine learning, which effectively compensates for the shortcomings of a single AI algorithm and improves the prediction performance.

As shown in Figure 1, certain internal stress generates in the manufacturing process of steel plates [7]. The magnitude of internal stress plays a key role in the application of steel products and the effect on final products. The internal stress of steel plates is unqualified, which may cause deformation and cracks of steel plates. Detecting unqualified steel plates and performing cold straightening is the key to improving the production quality of steel plates. The accuracy of whether steel plates need cold straightening cannot be guaranteed if it is only judged by workers' experience and naked eyes. Also, the effect is not ideal because it is usually affected by the status and quality of workers. These problems can be avoided by using AI technologies to make decisions on cold straightening of steel plates. The steel plate cold straightening auxiliary decision-making algorithm aims to predict whether the internal stress of steel products is qualified and cold straightening is necessary, which may lead to a more stable and accurate judgment on whether cold straightening is necessary in the steel plate production process.

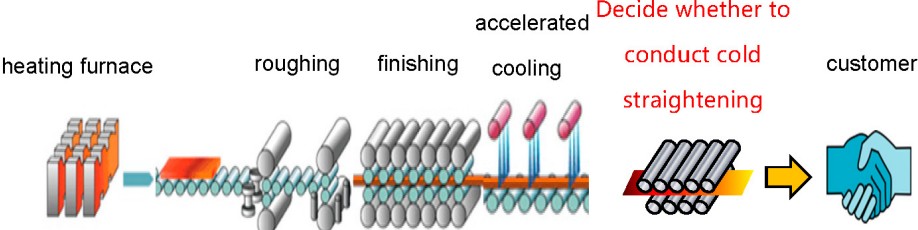

**Figure 1.** Steel plate manufacturing process.

However, traditional single machine learning algorithms cannot adapt well to different types of steel plates (including steel plates of different sizes and different physical compositions, as their production environment and corresponding data can be very different) and production processes in actual production because they often fall into local optimum easily, and the adaptability of the model is limited [8]. Because a single kind of classical AI algorithm has different capabilities to process data sets with different characteristics, there is also a bias towards the data processing capabilities of different sizes and types of steel plates. Using the multi-classifier ensemble learning method can make up for the shortcomings of different classifiers and improve the final classification prediction effect. At present, it is an urgent and common problem in the steel production industry to propose a steel plate cold straightening auxiliary decision-making method with higher judgment accuracy and stronger model adaptability [9].

Ensemble learning [10] is a common statistical learning method, which is widely applied and effective. Ensemble learning usually achieves significantly better generalization performance than a single learner by combining multiple learners. In the classification problem, it enhances the performance of classification by changing the weights of training samples, learning multiple classifiers and combining these classifiers linearly.

Based on the above industrial background, this project adopts the strategy of ensemble learning to propose a new type of steel plate cold straightening auxiliary decision-making method. With the data of rough rolling [11], finishing [12] and steel plates that have not been cold straightened [13] after accelerated cold treatment [14], this project analyzes factors which play a pivotal role in the internal stress and uses ensemble learning methods to build a strong machine learning method which is able to predict if steel plate cold straightening is necessary.

By predicting the subplate that needs to be cold straightened more effectively, this model reduces the cold straightening rate, the customer quality objections caused by residual stress and is applied to improving the yield of steel products.

## 2. Introduction to the Steel Plate Cold Straightening Auxiliary Decision-Making Data

We used SIEMENSE PLC in the production of steel plates, which used snap7 technology to collect production data. In the cold straightening stage of steel plates, a patented technology that automatically recognizes the FQC image content of thick plates to obtain

the corresponding data parameters in real time is used. Finally, the real-time database can be obtained by means of processing data by the data collection gateway.

The specific data acquisition process is shown in Figure 2.

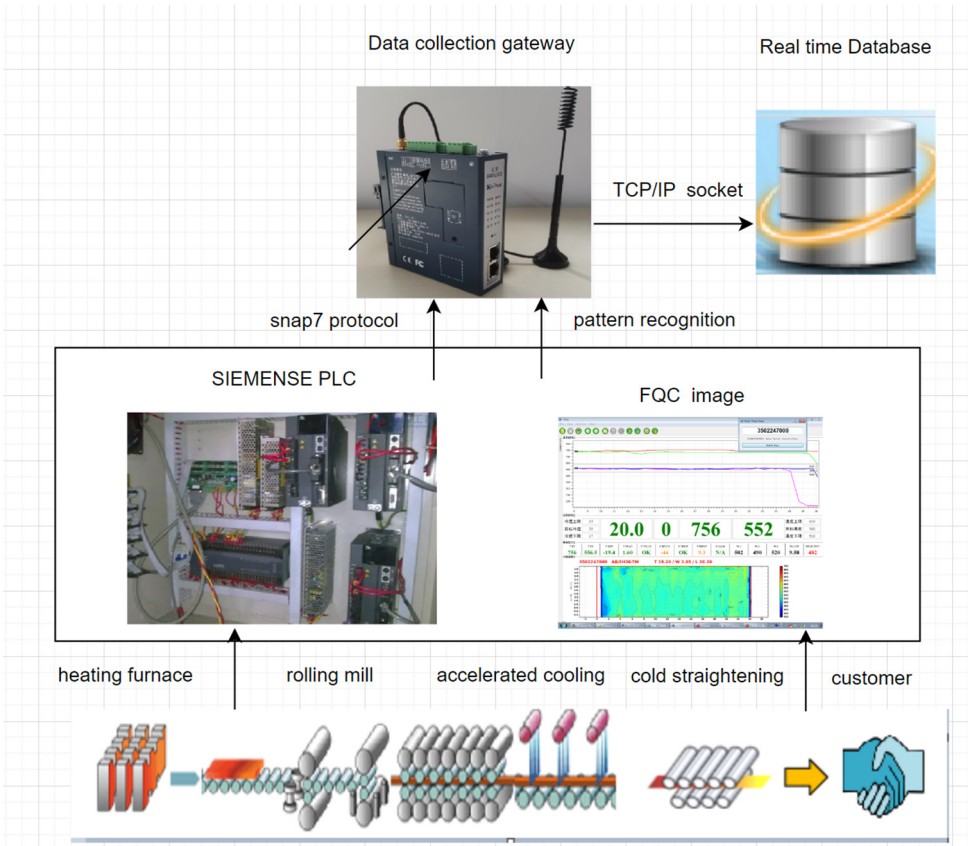

**Figure 2.** Dataset acquisition process.

We selected Baosteel's steel plates production data as the data source of this project. Since many of Baosteel's subplates from May to July 2020 were cold straightened [15], this paper selects the data of subplates that have not been cold straightened from January to April 2020 as the subject in this study. The experiment is aimed at the steel plate production process and purposefully slices steel plates whose uniformity of preheating and sizes are different; the experimental data of the study is composed of factors about rolling thickness, temperature, torque, bending force at stages of rolling mill and finishing and attributes like the production intervals, whether the production happens in a single furnace.

The main influencing factors for the internal stress in the steel plate and whether cold straightening is required are summarized in the experimental data, which is shown in Table 1 below.

**Table 1.** List of properties that may affect the internal stress in the steel plate and whether it is cold straightened or not.

| Variable Name | Physical Meaning |
| --- | --- |
| PH_EQUALITY | Uniformity of the preheating section |
| LENGTH | Subplate length |
| WIDTH | Subplate width |
| THICKNESS | Subplate thickness |
| PRE_EXITTHICKNESS_MAX_RM | The maximum value of the target value of the rollout thickness at the roughing stage |
| PRE_EXITTHICKNESS_MEAN_RM | The average of the target values for the roll thickness at the roughing stage |

**Table 1.** *Cont.*

| Variable Name | Physical Meaning |
| --- | --- |
| MEAS_TORQUETOP_MAX_RM | The maximum value of the torque measurement at the roughing stage |
| MEAS_TORQUETOP_MEAN_RM | The average of the torque measurements at the roughing stage |
| PRE_BENDINGFORCE_MAX_FM | The maximum value of the target value of the bending roll force at the finishing stage |
| PRE_BENDINGFORCE_MEAN_FM | The average value of the target value of the bending roll force at the finishing stage |
| MEAS_BENDINGFORCE_RANGE_FM | Change in bending roll force measurement at the finishing stage |
| PRE_EXITTEMPERATURE_MAX_FM | The maximum value of the target value of the outlet temperature at the finishing stage |
| PRE_EXITTEMPERATURE_MEAN_FM | The average of the target values for the rolling the temperature at the finishing stage |
| PRE_EXITTEMPERATURE_RANGE_FM | Change in target value of outgoing temperature at the finishing stage |
| ENTRYTEMP_MAX_RM | The maximum temperature before rolling at the roughing stage |
| ENTRYTEMP_MEAN_RM | The average of the pre-rolling temperatures at the roughing stage |
| ENTRYTEMP_RANGE_RM | Change in temperature before rolling at the roughing stage |
| IN_EVENT | The interval is fewer than 7 days and it is produced in a single furnace |
| EVENT | The time interval is greater than 7 days and it is produced in a single furnace |
| MULTI_IN_EVENT | The interval is fewer than 7 days and it is produced in multiple furnaces |
| MULTI_EVENT | The time interval is greater than 7 days and it is multi-furnace production |

## 3. Auxiliary Decision-Making System for Steel Plate Cold Straightening Based on Multi-Machine Learning

In the steel plate production and application, a single machine learning algorithm is usually used. However, a single learning algorithm lacks the ability to adapt to different steel plate types and manufacturing, which may lead to poor learning effects. As a result, the use of ensemble learning methods is an important strategy to effectively improve the accuracy and stability of learning.

Among numerous ensemble learning algorithms [16], the AdaBoost [17] algorithm, whose main strategy is weighted voting, is one of the most representative methods. Moreover, it follows the rule that the minority is subordinate to the majority. The AdaBoost algorithm constantly updates and optimizes the weight of each underlying algorithm through adequate training. Finally, the weight of the high-accuracy classifier is improved and the weight of the low-accuracy classifier is reduced. Consequently, the accuracy of the final integration decision is increased and the performance of the algorithm is improved during the voting process.

The rough set is an important theory for the reasoning of uncertain problems [18]. Data attribute reduction based on the rough set is a widely recognized method in dimensionality reduction of high-dimensional data [19]. This method which has been widely applied to various types of industrial data mining during recent years can be used to eliminate irrelevant attributes and reduce model complexity under basically the same decision and classification ability. The reduction result of taking the rough set is more objective and effective because it is not necessary to use any prior attribute distribution information. In this paper, a steel plate cold straightening auxiliary decision-making system based on multi-machine learning competitive strategy (MMLCS) is proposed. In addition, the rough set method is applied to blending the idea of ensemble learning and simplifying the

attributes of multiple attribute factors that affect the magnitude of the internal stress and decide whether cold straightening is required in the steel plate. At the same time, the idea of ensemble learning is fused, and a multi-machine learning competitive strategy (MMLCS) is proposed.

The industrial data mining algorithm model of the multi-machine learning competitive strategy is shown in Figure 3. The corresponding algorithmic framework of this model is as follows:

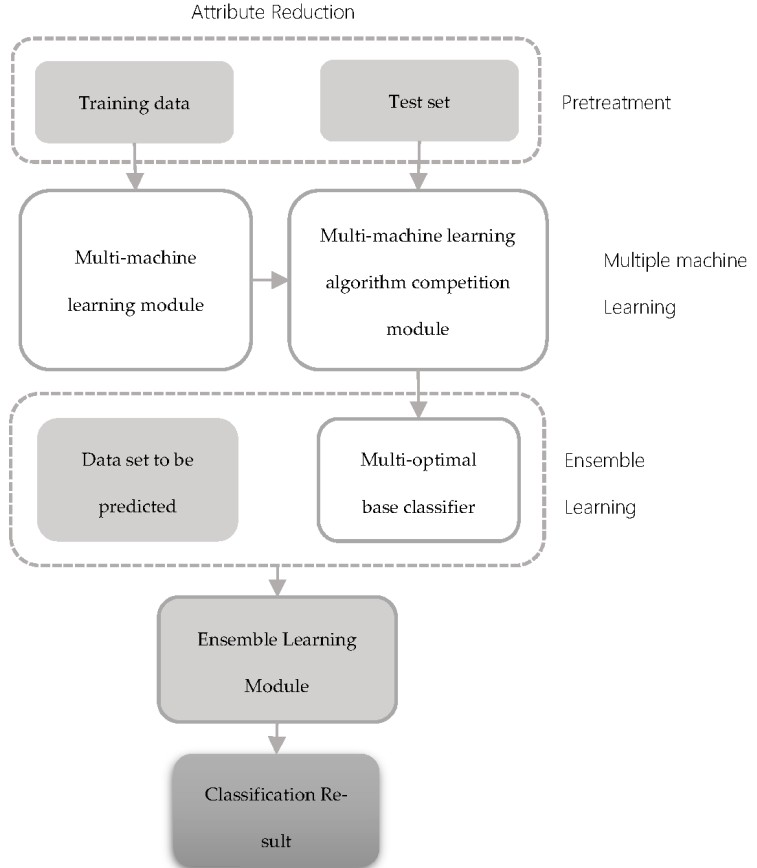

**Figure 3.** Algorithmic framework of multi-machine learning competitive strategy.

This model includes data reduction modules (containing data cleaning, data discretization, rough set), multi-machine competition module, ensemble learning, and other main modules. The learning process was as follows: First, training data set attribute reduction should be done. Then, the result should be imported into the multi-machine competition module. In the multi-machine competition module, $M$ classifiers are trained through a competitive strategy. $N$ optimal classifiers ($N < M$, $N$ is an odd number) are finally obtained and saved into the multi-optimal classifier module. In the end, in the ensemble learning module, a multi-optimal classifier is extracted, predicting the actual steel plate according to the integration strategy and calculating the optimal prediction results.

The following subsections will introduce some of the important modules involved in the model in detail.

### 3.1. Data Attribute Reduction

This cold straightening data set can be regarded as a knowledge expression system $IS = (U, R, V, f)$, wherein, $U = \{x_1\ x_2 \dots x_i\}$ represents the non-empty finite set of cold straightening data record objects(tuples), where the $x_i$ is the *No.i* cold straightening record(tuple). $R = C \cup D$ is the set of all attributes of the cold straightening data, divided into two subsets that do not intersect, namely conditional attribute $C$ and decision attribute $D$, conditional attribute $C$ includes all the attributes mentioned in Table 1, and the domain

of decision attribute $D$: $D \subset \{1, -1\}$, 1 means that cold straightening occurs, $-1$ means that no cold straightening occurs. $V$: The set of values for conditional attribute $C$, and $V_a$ is the domain of the property. $f$: $U \times R \rightarrow V$ is an information function that assigns a value to any property of $a \in R$ corresponding to any object $x \in U$, namely $f_a(x) \in V_a$. The Cold Straightening Prediction Information System is a large, complex and diverse data collection containing many redundant attributes. To obtain a simplified decision table, further processing is required.

This project uses a rough set reduction method, and the reduction algorithm is introduced in the Algorithms 1:

---

**Algorithms 1** RS_Reduction

---

Input: Cold Straightening Forecast Information System $IS = (U, C, D, V, f)$

---

Output: $IS^* = (U, C^*, D, V, f)$ after attribute reduction, wherein $C^* \subseteq C$
For the algorithm steps, check Reference [20].

---

### 3.2. Multi-Machine Competition and Integration Modules

The classifier race process in multiple machine learning competitions is shown in Figure 4. Firstly, input the initialized training data, and use the rough set to reduce attributes. The simplified data is used to learn and train $M$ different algorithms, with $M$ different knowledge produced after learning. The test data is then applied to competing for $M$ knowledge. Finally, $N$ classifiers with the highest prediction accuracy are outputted and saved into the optimal classifier module.

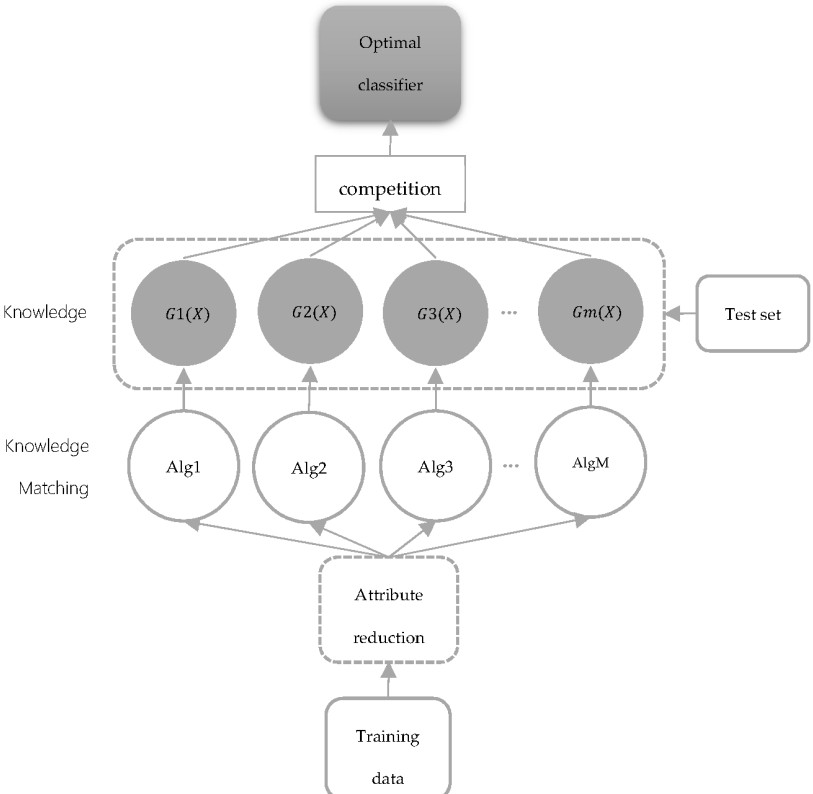

**Figure 4.** Network structure of the classifier.

The key algorithmic steps in the Multiple Machine Learning Competition module are as follows:

---

**Algorithms 2** MML_competition

---

Input: *M* alternative algorithm sets *X*,

---

$// \ X = \left\{ \text{Algorithm}^1(\ ), \text{Algorithm}^2(\ ), \cdots \text{Algorithm}^M(\ ) \right\}$

Training data set: *set*1, Test data set: *set*2

$// \ set1$ and *set*2 are data sets with the same data structure as *IS*

Output: *N* optimal classifiers set *Y*

$set1' = \text{RS\_Reduction}(set1); \ set2' = \text{RS\_Reduction}(set2)$

for (i=1; i< *M*; i++)

    {

    Substituting $set1'$ into Algorithm$^i$( ) for machine learning;

    *//learning process is omitted*

    Import the test data $set2'$ into the learned algorithm Alg$^i$( ) to obtain the accuracy of the prediction, recorded as $a_i$.

    *//the calculation formula:*

$$a_i = \left( \frac{\sum_{j=1}^{|set2'|} \frac{\text{Alg}^i\left(c^j\right) \ \text{iff} \ d^j}{d^j}}{|set2'|} \right)$$

    Wherein $c^j \in C_{set2'}$, $d^j \in D_{set2'}$, $|set2'|$ represents the number of sets, $\text{Alg}^i\left(c^j\right)$ represents the conditional property of the *j* record *Cj* is substituted into algorithm *i*,

    }

for (I = l; I < *N*; i++)

    {

    for (j = l; j< *M*; j++)

    {

        Import the classifier Alg$^i$( ) with the best accuracy in this cycle and its accuracy $a_i$ into *Y*, and delete Alg$^i$( ) as well as *M*=*M*−1 in *X*

    }

    }

---

After obtaining the optimal classifier set *Y*, the method of ensemble learning can be applied to making predictions for cold straightening. The integration strategy uses accuracy and variance as weights, respectively. Accuracy is used as a weight for prediction first. When the outcome is uncertain, variance is used as an alternative weight for prediction.

The integration strategy is: At the initial learning stage, the prediction data set is used to learn the *N* algorithms generated in the multi-machine learning competition module, and the prediction accuracy of the *N* algorithms is used as the respective weights. Secondly, the prediction results in *N* algorithms are divided into two groups according to the occurrence of cold straightening and non-occurrence of cold straightening. The weighted prediction accuracy as well as the number of algorithms in the group are counted separately. If the number of algorithms in the cold straightening group is greater than that in the no cold straightening group, and the sum of the prediction accuracy of the cold straightening group algorithm is greater than that of the no cold correction group algorithm, the prediction result is cold straightening; Alternatively, if the number of algorithms in the cold straightening group is fewer than that in the no cold straightening group, and the sum of the prediction accuracy of the cold straightening group algorithm is also fewer than that of the no cold straightening group algorithm, the prediction result is no cold straightening.

Otherwise, count the accuracy of the algorithm for the first five times separately, calculate and compare the variance of each algorithm in the group, and the smaller variance is the output of this forecast.

The specific algorithm pseudo code mentioned above is as follows:

| **Algorithms 3** MML_decision |
|---|
| Input: Prediction data set: *set*3; Optimal classifiers set: $Y$; |
| Output: Final prediction result <br> Initialize the variable *result*; <br> for (i = 1; I < $N$; i++) <br>     { <br> *//Learn the N algorithms in set Y separately with the prediction dataset set3.* <br> *//1 means that cold straightening occurs, −1 means that no cold straightening occurs;* <br>     Calculate $R_i = \|a_i \times Y_i\|$; and ResultList += $a_i$. <br>     *//Count the prediction result $a_i$ of each algorithm separately, and assign the accuracy $Y_i$ of the algorithm in Y as a weight to $a_i$. The calculation result $R_i$ is added to the ResultList.* <br>     Coldstraightening = group(ResultList) <br>     *// Group N results in the ResultList according to cold straightening and no cold straightening, and calculate the prediction accuracy of each group.* <br>     } <br>     ColdstraighteningCount = Count(Coldstraightening) <br>     *// Count the number of algorithms that coldstraightening occurs andcoldstrighteningdoes not occur respectively.* <br> If <br>     {(ColdstraighteningCount(1) > ColdstraighteningCount(−1) && (Coldstraightening(1) > Coldstraightening(−1)) <br>   *Result* =1 <br> Else if <br>     (ColdstraighteningCount(1)< ColdstraighteningCount(−1)&&(Coldstraightening(1)< Coldstraightening(−1)) <br>   *Result* = −1 <br>     } <br> Else <br>     { <br>     precisionList = Var($Y$) <br>     *// Compute the variance of last five accuracy of each algorithm by group, and Y has the prediction accuracy of N algorithms each time.* <br>     If(precisionList(1) > precisionList(−1)) <br>   *Result* = −1 <br>     Else <br>   *Result* = 1 <br>     } <br> Output the prediction result: *Result*. <br> *// The value of the prediction result is the magnitude of the average which represents whether coldstraightening occurs* <br> Update $Y$ |

## 4. Experimental Results and Analysis of the Steel Plate Cold Straightening Auxiliary Decision-Making

### 4.1. Experimental Environment

This experiment uses the Windows10 system. Matlab7.1 and GrADS1.9 are used as the experimental platform. In the algorithmic framework of the multi-machine learning competitive strategy, the classification algorithm uses the SVM classification algorithm, C4.5 decision tree algorithm [21], random forest [22], naïve Bayes classifier [23], ANN artificial neural network classification algorithm [24] and k-near neighbor classification algorithm [25].

### 4.2. Experimental Data Preparation

As shown in Table 2, since many of Baosteel's subplates from May to July 2020 were cold straightened [15], the data of subplates that were not cold straightened from January to April was selected as the subject in this study.

**Table 2.** Current situation of cold straightening from January to July 2020.

| Cold Straightening | Jan | Feb | Mar | Apr | May | Jun | Jul |
|---|---|---|---|---|---|---|---|
| TRUE | 490 | 353 | 215 | 151 | 503 | 880 | 178 |
| FALSE | 630 | 693 | 326 | 193 | 114 | 94 | 42 |
| Sum | 1120 | 1046 | 541 | 344 | 617 | 974 | 220 |

The reduced attributes of the data are shown in Table 3.

**Table 3.** Finally yield nine attribute variables that contribute to the key influencing factors of cold straightening.

| Mark | Variable Name | Physical Meaning |
|---|---|---|
| X1 | PH_EQUALITY | Uniformity of the preheating section |
| X2 | LENGTH | Subplate length |
| X3 | PRE_EXITTHICKNESS_MAX_RM | The maximum value of the target value of the rollout thickness at the roughing stage |
| X4 | MEAS_TORQUETOP_MEAN_RM | The average of the torque measurements at the roughing stage |
| X5 | PRE_BENDINGFORCE_MEAN_FM | The average of the target value of the bending roll force at the finishing stage |
| X6 | MEAS_BENDINGFORCE_RANGE_FM | Changes in the measurement of the bending roll force at the finishing stage |
| X7 | PRE_EXITTEMPERATURE_RANGE_FM | Changes in the target value of the rolling temperature at the finishing stage |
| X8 | ENTRYTEMP_RANGE_RM | Changes in the temperature before rolling at the roughing stage |
| X9 | EVENT | The time interval is greater than 7 days and it is produced in a single furnace |

The model was cross-validated by the use of data from the first four months of 2020. The selection strategy of training data and test data is shown in Figure 5 below:

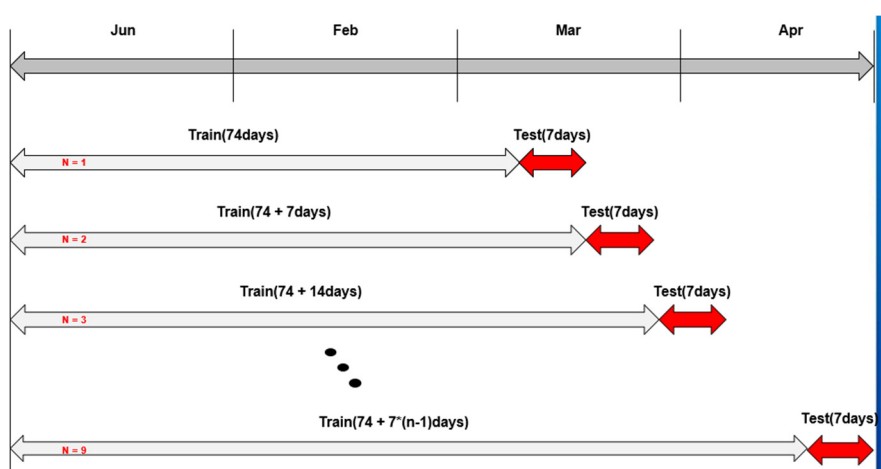

**Figure 5.** Verify the rules of data selection.

The results of the cross-validation are shown in Table 4.

**Table 4.** The results of the cross-validation.

| N | Train | Test | Models | Good | Defect |
|---|---|---|---|---|---|
| 1 | $1/2 - 3/13$ | $3/14 - 3/20$ | $-8.5461496 + 0.0181917 \times X1 - 0.020788 \times X2 - 7.268501 \times X3 + 0.0006369 \times X4 + 0.0255786 \times X5 + 0.0020323 \times X6 + 0.0540281 \times X7 - 7.0883774 \times X8 + 4.063 \times X9$ | 98.8% | 67% |

**Table 4.** *Cont.*

| N | Train | Test | Models | Good | Defect |
|---|---|---|---|---|---|
| 2 | 1/2 − 3/20 | 3/21 − 3/26 | $-7.659139 + 0.017657 \times X1 - 0.032719 \times X2 - 18.797067 \times X3 + 0.001265 \times X4 + 0.029409 \times X5 + 0.002721 \times X6 + 0.048697 \times X7 - 5.854059 \times X8 + 3.87 \times X9$ | 100% | (100%) |
| 3 | 1/2 − 3/26 | 4/2 − 4/3 | $-7.39 + 0.02057 \times X1 - 0.04098 \times X2 - 17.83 \times X3 + 0.001469 \times X4 + 0.03021 \times X5 + 0.003151 \times X6 + 0.05464 \times X7 - 6.939 \times X8 + 3.78 \times X9$ | 97.8% | 100% |
| 4 | 1/2 − 4/3 | 4/4 − 4/10 | $-7.05 + 0.02081 \times X1 - 0.04119 \times X2 - 18.31 \times X3 + 0.001401 \times X4 + 0.0294 \times X5 + 0.003752 \times X6 + 0.05509 \times X7 - 7.088 \times X8 + 4.162 \times X9$ | 100% | (100%) |
| 5 | 1/2 − 4/10 | 4/11 − 4/15 | $-7.06 + 0.02058 \times X1 - 0.04094 \times X2 - 18.5 \times X3 + 0.001393 \times X4 + 0.02949 \times X5 + 0.003744 \times X6 + 0.05492 \times X7 - 7.035 \times X8 + 4.083 \times X9$ | 100% | (100%) |
| 6 | 1/2 − 4/15 | 4/18 − 4/24 | $-6.98 + 0.02051 \times X1 - 0.04155 \times X2 - 18.56 \times X3 + 0.001394 \times X4 + 0.02962 \times X5 + 0.003759 \times X6 + 0.05556 \times X7 - 7.135 \times X8 + 3.92 \times X9$ | 100% | (100%) |
| 7 | 1/2 − 4/24 | 4/25 − 4/30 | $-7.07 + 0.02024 \times X1 - 0.04141 \times X2 - 18.62 \times X3 + 0.001404 \times X4 + 0.02998 \times X5 + 0.003811 \times X6 + 0.05555 \times X7 - 7.113 \times X8 + 3.988 \times X9$ | 100% | (100%) |

Then, we choose the optimal model:

$$f(x) = -19.93 + 0.01793 \times X1 + 0.0002346 \times X2 - 0.03998 \times X3 + 0.002326 \times X4 + 0.03703 \times X5 - 0.01078 \times X6 + 0.02663 \times X7 + 0.004614 \times X8 + 3.779 \times X9$$

In the experiment, the calculation formulas of the classifier's good hit rate and defective hit rate are as follows:

$$Good\ product\ hit\ rate = \frac{Predict\ the\ number\ of\ good\ products}{The\ actual\ number\ of\ good\ products} \times 100\%$$
$$Defective\ hit\ rate = \frac{Predict\ the\ number\ of\ defective}{The\ actual\ number\ of\ defective} \times 100\%$$

As shown in Table 5, by using the data from April to June 2020, the model predicts that the good hit rate is 97.9% and the defective hit rate is 90.9%.

**Table 5.** Model accuracy confusion matrix.

| | | Actual | | |
|---|---|---|---|---|
| | | **Normal** | **Defective** | **Sum** |
| Predict | normal | 281 (97.9%) | 1 (9.1%) | 282 (94.6%) |
| | defective | 6 (2.1%) | 10 (90.9%) | 16 (5.4%) |
| | sum | 287 (100%) | 11 (100%) | 298 (100%) |

The experimental results of this application indicate that the hit rate of good products is 97.9% and the hit rate of defective products is 90.9% by adopting the cold straightening prediction algorithm of steel plates based on the multi-machine learning competition

strategy to predict whether the subplates should be cold straightened. This method improves the product quality of subplates effectively. Not only does this application solve the decision-making problem of whether subplates should be cold straightened or not, but also it reduces the costs of cold straightening of all products and the risk of product defects caused by cold straightening. Consequently, this application has important meanings in practical engineering application.

In order to compare the advantages and disadvantages of the MMLCS algorithm in this paper with other classical algorithms, we set up a comparative experiment. A contrast experiment which is used to predict the yield of steel plates is performed in the view of the steel plate cold straightening auxiliary decision algorithm based on the multi-machine learning competition strategy (MMLCS) and other classical machine learning decision algorithms (SVM, decision tree, and naïve Bayes algorithm as examples).

The experimental results are shown in Figure 6. In terms of the good product hit rate, naïve Bayes gets the highest hit rate (93.10%) among three classical machine learning decision algorithms. In comparison, the MMLCS gets 97.90% in the good product hit rate. In terms of the defective hit rate, naïve Bayes gets the best hit rate (88.60%) among the classical machine learning decision algorithms in the experiment, while MMLCS gets 90.90% in the defective hit rate, which achieves a better performance. In conclusion, the MMLCS steel plate cold straightening decision algorithm in this paper is better than the other three classic machine learning decision algorithms in terms of the good product hit rate and the defective hit rate.

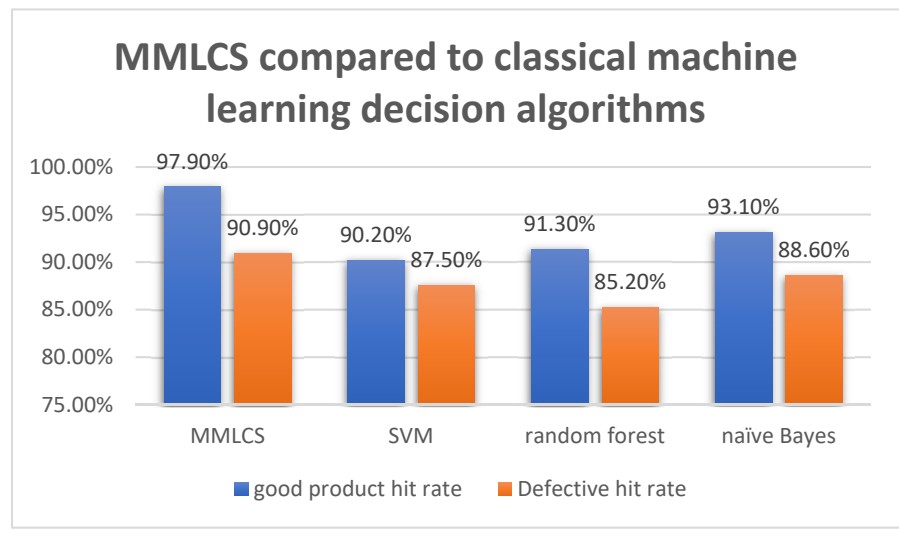

**Figure 6.** MMLCS compared to classical machine learning decision algorithms.

## 5. Conclusions

This paper aims at solving the problem of whether cold straightening is required due to the influence of internal stress in the steel plate production process. To reduce the production cost of the steel plate and improve the qualified rate, an auxiliary decision-making method for cold straightening of steel plates is proposed. The algorithm adopts the multi-machine learning competition strategy. First of all, in order to reduce data dimension, the attributes of the conditional factors, which affect the internal stress and whether steel plates required to be cold straightened, should be reduced. Secondly, the competition of training multiple different learners on the data set produces the optimal base classifier. Finally, the final classifier is generated by training the weights on the optimal base classifier and combining it with a centralized strategy. Good application results are conducive to reducing the production cost of the steel plate and improving the qualified rate. In practical engineering, this research method can also form a series of cold straightening prediction model libraries related to actual model products and achieve the effect of intelligent production by establishing prediction models of subplates with

different types. Additionally, product qualities are effectively improved and production costs are significantly reduced. In sum, the method proposed in this paper is well worth being popularized.

**Author Contributions:** Conceptualization, Z.-H.D., R.-H.W. and J.-H.G.; methodology, Z.-H.D.; software, R.-H.W. and J.-H.G.; validation, Z.-H.D., R.-H.W. and J.-H.G.; formal analysis, Z.-H.D.; investigation, Z.-H.D.; resources, R.-H.W.; data curation, Z.-H.D.; writing—original draft preparation, Z.-H.D.; writing—review and editing, R.-H.W.; visualization, R.-H.W.; supervision, Z.-H.D.; project administration, R.-H.W. All authors have read and agreed to the published version of the manuscript.

**Funding:** This research received no external funding.

**Institutional Review Board Statement:** Not applicable.

**Informed Consent Statement:** Not applicable.

**Data Availability Statement:** Not applicable.

**Conflicts of Interest:** The authors declare no conflict of interest.

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
