# Peer review of "Auxiliary Decision-Making System for Steel Plate Cold Straightening Based on Multi-Machine Learning Competition Strategies"

_applsci, doi:10.3390/app122211473_

Round 1

Reviewer 1 Report

Thanks for the performed work. Please find attached my comments for the manuscript.

Author Response

Thanks for the comments and suggestions to the article.

Reviewer 2 Report

It is unclear what train:test split ratio was used during classification? Results must be provided considering different holdout % and holdout validation approaches. Refer to this article to understand the holdout validation approach. ‘A Bayesian Optimized Discriminant Analysis Model for Condition Monitoring of Face Milling Cutter Using Vibration Datasets’. In addition to this, results of k-fold cross-validation must be provided.

Include an actual picture of the experimentation setup. In addition clearly explain the process of data collection, instrumentation used, etc.

What about classification using a blind dataset (No labels)? Classification results must be provided considering the blind dataset (No labels). Apply the trained model to classify blind datasets. Refer to “Figure 8: Framework for classification of blind data” from the following article ‘Application of Bayesian Family Classifiers for Cutting Tool Inserts Health Monitoring on CNC Milling’.

Refer some recent references for bagging and boosting such as ‘Design of Bagged Tree Ensemble for carbide coated inserts fault diagnosis’, ‘Supervision of Carbide Tool Condition by Training of Vibration-based Statistical Model using Boosted Trees Ensemble’, ‘Cutting Tool Condition Monitoring using a Deep Learning-based Artificial Neural Network,’ ‘Multi-Point Face Milling Tool Condition Monitoring Through Vibration Spectrogram and LSTM-Autoencoder,’

How to deal with the diversity between the data distributions of present and future moments? Traditional ML algorithms can only resolve classification issues within the same data distributions. In predicting various parameters, what would be the critical steps of generalizing to unknown moments?

How to predict future faults with the collected raw signal from the present configuration of defined levels of different factors?

The grammar is improvable in a few places. There must be thorough proofreading of the paper.

Author Response

(The authors gave the same response as above.)

Round 2

Reviewer 1 Report

Thanks for addressing my comments. Please find attached my responses. Some comments need to be further addressed.

Author Response

Thanks for the reviewing and advice for the manuscript. 

I have revised and improved the manuscript according to the comments in response. And made some further clarification to the responses. 
I have marked the main changes using the MS word’s “Track Changes” function to be easily viewed.

Reviewer 2 Report

The authors have tried addressing my comments, and the paper may be accepted.

Author Response

Thanks for the reviewing and suggestions again.